# Towards Capturing the Temporal Dynamics for Trajectory Prediction: a Coarse-to-Fine Approach

**Xiaosong Jia**[1,2], **Li Chen**[2], **Penghao Wu**[1,2], **Jia Zeng**[1,2], **Junchi Yan**[1,2*], **Hongyang Li**[1,2], **Yu Qiao**[2]

[*]corresponding author

[1] Shanghai Jiao Tong University [2] Shanghai AI Laboratory

{jiaxiaosong, wupenghaocraig, jia.zeng, yanjunchi}@sjtu.edu.cn

{lichen, lihongyang, qiaoyu}@pjlab.org.cn

**Abstract:** Trajectory prediction is one of the basic tasks in the autonomous driving field, which aims to predict the future position of other agents around the ego vehicle so that a safe yet efficient driving plan could be generated in the downstream module. Recently, deep learning based methods dominate the field. State-of-the-art (SOTA) methods usually follow an encoder-decoder paradigm. Specifically, the encoder is responsible for extracting information from agents' history states and HD-Map and providing a representation vector for each agent. Taking these vectors as input, the decoder predicts multi-step future positions for each agent, which is usually accomplished by a single multi-layer perceptron (MLP) to directly output a Tx2 tensor. Though models with adoptation of MLP decoder have dominated the leaderboard of multiple datasets, 'the elephant in the room is that the temporal correlation among future time-steps is ignored since there is no direct relation among output neurons of a MLP. In this work, we examine this design choice and investigate several ways to apply the temporal inductive bias into the generation of future trajectories on top of a SOTA encoder. We find that simply using autoregressive RNN to generate future positions would lead to significant performance drop even with techniques such as history highway and teacher forcing. Instead, taking scratch trajectories generated by MLP as input, an additional refinement module based on structures with temporal prior such as RNN or 1D-CNN could remarkably boost the accuracy. Furthermore, we examine several objective functions to emphasize the temporal priors. By the combination of aforementioned techniques to introduce the temporal prior, we improve the top-ranked method's performance by a large margin and achieve SOTA result on the Waymo Open Motion Challenge.

**Keywords:** Autonomous Driving, Trajectory Prediction, Temporal Correlation

## 1 Introduction

Trajectory prediction aims to predict agents' positions in the next several seconds, which is one of the essential tasks for the autonomous driving. Recently, deep learning based methods have dominated the field and they usually follow an encoder-decoder paradigm [1, 2]. The encoder takes all agents' history states (position, velocity, heading, *etc.*) and HD-Map (lanes, road lines, traffic lights, *etc.*) as inputs and outputs a representation vector for each agent. The decoder then generates future trajectories for the agents based on these vectors. *We observe that current top-ranked methods [3, 4, 5, 6, 7, 8, 9, 10, 11, 12, 13, 14] on multiple datasets*[1] *all adopt an MLP as the decoder to output a* $T \times 2$ *tensor and then reshape it into the final prediction.*

---

[1]The investigation is based on their corresponding public leadrboard as of the date of the submission. DCMS [3] and HiVT [4] on Argoverse [15], MultiPath++ [5], SceneTransformer [6], DenseTNT [7], and HDGT [8] on Waymo Open Dataset [9], Multimodel Tranformer [10], GOHOME [11] and THOMAS [12] on INTERACTION [16], PGP [13] and LaPred++ [14] on nuScenes [17].

6th Conference on Robot Learning (CoRL 2022), Auckland, New Zealand.

Though the MLP design achieves satisfying performance empirically, we point out that it ignores the temporal correlation among the future time-steps. Mathematically, we denote the hidden representation vector of an agent as $\mathbf{H} \in \mathbb{R}^H$ and its predicted future trajectory as $\tau \in \mathbb{R}^{T \times 2}$ where $T$ indicates the length of prediction horizon and 2 indicates the 2D coordinate. The MLP accomplishes the mapping:

$$\tau = \mathbf{W}_2 \delta(\mathbf{W}_1 \mathbf{H}).\text{reshape}(T, 2), \tag{1}$$

where $\mathbf{W}_1 \in \mathbb{R}^{H' \times H}$ and $\mathbf{W}_2 \in \mathbb{R}^{(2T) \times H'}$ are weight matrices, and $\delta(\cdot)$ is a non-linear function. We can find that every two rows of $\mathbf{W}_2$ contains the weights to generate the position for one specific time-step. Since aforementioned SOTA works adopt either per-step negative log likelihood (NLL) loss or per-step mean squared error (MSE) loss, the back-propagation updating for every two rows of $\mathbf{W}_2$ is independent. Such design is based on the assumption of conditional independence among future time-steps. However, the assumption is untenable in real-world scenarios and thus could be harmful for the prediction.

In this work, we aim to discuss about 'the elephant in the room' and explore several approaches to model the temporal correlation for future time-steps on top of a SOTA encoder. We find that simply replacing the MLP with an autoregressive RNN leads to significant performance drop, even with techniques such as teacher forcing [18] or history highway [19] to alleviate its drawbacks of long-term dependencies and cumulative error. Inspired by the success of refine modules in the deep learning based methods [20, 21, 22, 23] which ease the optimization difficulty of a single decoder head, **we find that when first using an MLP to generate a scratch trajectory and then using a structure with temporal inductive bias (RNN/1D-CNN) to refine it, the SOTA model's performance could be boosted**. We also explore different training objectives to apply temporal priors on the output besides modifications of the neural network structure. We find that directly fitting the velocity degenerates the performance, while accumulating the velocity into the displacement (*i.e.*, calculating the integral of velocity over time) and still using the coordinates as the target would lead to better prediction. By combining these discovered effective techniques, we significantly boost the performance of a SOTA model on Waymo Open Motion Dataset, which verifies the benefit of utilizing the temporal prior.

In summary, the contributions of the paper are:

- We explore ways to incorporate the temporal structure in the decoder and propose an effective scratch-then-temporally-refine paradigm. By using an MLP and a temporal module in two steps respectively, it explicitly models the temporal correlation among future time steps without the burden of training an autoregressive RNN.

- We examine several objective functions to add temporal priors to the output. We report the notable improvements by first generating velocity and then accumulating temporally into coordinates to calculate per-step loss.

- We conduct thorough ablation study on the Waymo Open Motion Dataset to test different ways of introducing the temporal prior into the decoding process. With the combination of our effective attempts, the performance of a SOTA encoder is boosted by a large margin on Waymo Public Leaderboard.

We consider the correlation among future time steps as a non-negligible factor, which is not exploited by current SOTA methods. We hope the attempts and successful parts illustrated in this study could provide useful information for this line of study in the trajectory prediction community.

## 2 Related Works

Pioneering works of deep learning based trajectory prediction models usually adopt an RNN decoder to generate future trajectories in an autoregressive way. Social LSTM [24] applies an LSTM decoder after the LSTM encoder, and adopts the NLL loss for each time-step. Its extension, Social GAN [25], uses GAN [26] to encode different modalities and adopts per-step MSE loss. DESIRE [27] employs a conditional variational autoencoder to sample a diverse set of hypothetical modalities and adopts a GRU to generate future predictions. MFP [28] introduces a latent distribution to model the joint modality of multiple agents and utilizes an RNN for decoding as well. Mulitpath [29] leverages a fixed set of future state-sequence anchors that correspond to modes of the trajectory distribution and regresses offsets along with uncertainties with respect to anchor waypoints by RNN. IDE-Net [30]

applies additional loss to infer the interaction types in an unsupervised fashion. To capture the temporal dynamics, Trajecton++ [31] utilizes a GRU as the decoder to predict a bivariate Gaussian distribution over control actions. By applying dynamically-extended unicycle models over the predicted control signals, it is able to guarantee that its trajectory samples are dynamically feasible. DKM [32] designs an autoregressive differentiable dynamics layer to generate the control signal per-step.

Though it is intuitive to use an autoregressive RNN to generate the sequence of future positions due to its recurrent nature, modern methods with a concise MLP decoder outperform previous RNN-based ones and dominate multiple public motion prediction leaderboards, including Waymo [9], nuScenes [17], Argoverse [15] and INTERACTION [16]. [33] utilizes CNN to encode the scene and generates trajectories directly by an MLP, while LaneGCN [2] encodes the scene by four distinct graph convolutional network (GCN) layers. PGP [13] traverses the lane graph to incorporate the lane topology while LaPred [14] applies attention mechanism on lanes. HiVT [4] designs a hierarchical way to aggregate the local and global information efficiently. SceneTransformer [6] encodes the spatial and temporal information in a factorized way. HDGT [8] models the driving scene as a heterogeneous graph. TPCN [34] and its extension DCMS [3] encodes the scene in the form of point cloud and apply an MLP on the pooled instance vector to generate future trajectories. VectorNet [1] encodes both the agents and lanes as vectors on a global graph and decodes the future trajectories based on the corresponding agents' vectors with an MLP. Its extension TNT [35] and DenseTNT [7] generate trajectories in a two-stage way: the stage one aims to predict the target/goal of agents and the stage two aims to complete the entire trajectories conditioned on the predicted end point by an MLP. GOHOME [11] and its extension THOMAS [12] output heatmaps for goals and then sample trajectories based on the selected goals. LaneRCNN [36] follows the goal-based prediction framework as well but they complete the trajectories with a polynomial prior and use an MLP to refine the polyline. In PLOP [37], means of the distribution of future coordinates are generated in polynomials of degree 4 of time. In the recent study - Multipath++ [5], they find that both polynomial representations and control signal representations lead to degenerated performance compared to directly regressing the coordinates.

Revisiting the existing literature, we can find that decoders and training objectives with different types of temporal structure are proposed. However, they are usually entangled with the encoder module and other settings, which makes it hard to draw a clear conclusion about the advantages and disadvantages of these designs. In this work, we examine different choice of applying temporal structure in the generation process of future trajectories while keep other settings such as the dataset, the encoder, the training schedule, and the hyperparameter the same for a fair comparison. To the best of our knowledge, we are the first to explore this design choice with the modern vector-based encoder on large-scale autonomous driving datasets.

## 3 Problem Formulation

Given history states $\{\mathbf{S}_{his}^i | i = 1, ..., N\}$ of $N$ agents such as their coordinate, velocity, and heading and their surrounding information $\mathbf{S}_{env}$ including lanes, traffic lights, *etc*., the goal is to predict their future positions in the next $T$ time-steps $\{\boldsymbol{\tau}^i \in \mathbb{R}^{T \times 2} | i = 1, ..., N\}$. Current SOTA methods usually follow the encoder-decoder paradigm. The encoder takes all agents' histories and environment information as inputs and outputs a representation vector $\mathbf{H}^i \in \mathbb{R}^H$ for each agent i:

$$\{\mathbf{H}_i | i = 1, ..., N\} = \text{Encoder}(\{\mathbf{S}_{his}^i | i = 1, ..., N\}, \mathbf{S}_{env}). \quad (2)$$

Conditioned on agents' hidden feature $\mathbf{H}_i$, the decoder aims to predict the distributions of their future trajectories respectively:

$$p(\tau_i | \mathbf{H}_i) = p(\mathbf{Y}_i^1, ..., \mathbf{Y}_i^T | \mathbf{H}_i), \quad i = 1, ..., N. \quad (3)$$

$\mathbf{Y}_i^t$ is agent $i$'s distribution of states at future time-step $t$.

As mentioned in Sec. 1, most SOTA works generate coordinates of different future time-steps with an MLP and adopt a step-wise MSE or NLL loss, which assumes that the distributions of states on different future time-steps are conditionally independent:

$$p(\mathbf{Y}_i^1, ..., \mathbf{Y}_i^T | \mathbf{H}_i) = \prod_{t=1}^{T} p(\mathbf{Y}_i^t | \mathbf{H}_i), \quad i = 1, ..., N. \quad (4)$$

We argue that this assumption ignores the temporal correlation among future time-steps, which could result in less consistent and physically infeasible trajectories. In this work, we explore two parts to apply the temporal priors: the mapping function $\phi(\mathbf{H}_i) = \hat{\tau}_i = \{\hat{\mathbf{Y}}_i^t | t = 1, ...T\}$ and the objective function $\mathcal{L}(\tau_i, \phi(\mathbf{H}_i))$.

**Experiments Setting:** We examine different decoders and objective functions on the validation set of Waymo Open Motion [9]. For all models, we adopt the same encoder module - HDGT [8], a top-ranked encoder on Waymo Leaderboard. All experiments are conducted with the same setting and hyperparameters. Please refer to Sec. 9 for more details. The following metrics are used: a) **minADE** (Minimum Average Distance Error): the minimum value of the Euclidean distance between the prediction and ground truth averaged by the prediction length $T$, for $K$ required predictions. b) **minFDE** (Minimum Final Distance Error): similar to minADE, despite that it only calculates the error at the final time-step $T$. c) **MR** (Missing Rate): the ratio of whether the Euclidean distance between the prediction and ground truth at the final time-step $T$ for all $K$ predictions is larger than 2 meters.

# 4 Temporal Enhanced Decoder

## 4.1 Autoregressive RNN

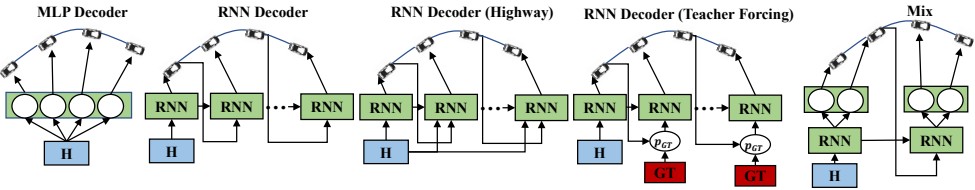

Figure 1: Illustration of the MLP, autoregressive RNN decoder, and their variants.

The autoregressive RNN predicts future trajectories in an iterative way with the assumption that the coordinate distribution of each time-step depends on that from previous time-steps (for notation brevity, we drop the subscript $i$ of agents in the following sections):

$$p(\mathbf{Y}^1, ..., \mathbf{Y}^T | \mathbf{H}) = \prod_{t=1}^{T} p(\mathbf{Y}^t | \mathbf{Y}^1, ...\mathbf{Y}^{t-1}, \mathbf{H}), \qquad (5)$$

The generation process is implemented by maintaining an internal state vector in RNN. Two commonly used RNN structures are LSTM [38] and GRU [39] and we adopt an LSTM here for its high capacity[2]. Meanwhile, there are two widely known drawbacks of RNN: the difficulty to capture long-term dependency due to the implicit past information retention mechanism [40] and the cumulative error [18]. As for the first problem, inspired by highway RNN [41, 42], we implement a variant of LSTM which takes the vector $\mathbf{H}$ as an additional input at each time-step so that the history information is always accessible. To

Table 1: Performance comparison of different decoder structure. H indicates the hightway technique and TF indicates the teacher forcing technique. Note that in Mix decoder, we adopt $l = 20$ which means autoregressing for 4 steps.

| Decoder | minADE↓ | minFDE↓ | MR↓ |
|---------|---------|---------|--------|
| MLP | **0.6056** | **1.2328** | **0.1723** |
| RNN | 0.7259 | 1.4903 | 0.2116 |
| RNN (H) | 0.7454 | 1.5438 | 0.2272 |
| RNN (TF) | 0.7179 | 1.4575 | 0.2053 |
| Mix | 0.7002 | 1.4620 | 0.2033 |

alleviate the cumulative error of autoregressive model, we adopt the teacher forcing with scheduled sampling strategy [18]. Specifically, at each time-step, the LSTM would have a possibility $p_{GT}$ to take the ground-truth (GT) state as input instead of the predicted one, and the possibility $p_{GT}$ decays as the training goes by. In this way, the model could be updated with less noise at the start of training while at the end it could learn to robustly do the prediction based on its previous output [18]. To futher explore the differences between the RNN and MLP decoder, we design a 'mix' of them. We

---

[2]Please refer to [38] for details about LSTM.

use an LSTM to autoregressively generates $l$ steps' representation vectors and then use an MLP to predict positions of $T/l$ time-steps with each step's vector as input.

In Figure 1, we give illustrations of the aforementioned decoders. Table 1 is the performance of the different decoder structures on Waymo Motion Dataset (validation set). We can conclude that:

- Autoregressive RNN performs worse than the MLP by a large margin, which explains why most SOTA methods adopt the latter one.

- The cumulative error is one cause of autoregressive RNN's bad performance. The teacher forcing technique could alleviate this problem, but its performance is still worse than MLP.

- Concatenating the history feature $\mathbf{H}$ with each time-step's state as input to RNN could not solve the long-term dependency problem. Indeed, it leads to worse performance than vanilla RNN. It might come from the difficulty of updating $\mathbf{H}$ for all $T$ time-steps as the input to RNN.

- The mix of the two structure has the performance between the MLP and RNN decoder. By reducing the autoregressive steps, the optimization becomes easier. However, it is still much worse than a concise MLP decoder.

In summary, we find that the inherent issues of autoregressive RNN make it hard to compete with the MLP decoder in the trajectory prediction task. We should explore other ways to apply the temporal inductive bias on the network structure.

## 4.2  Scratch-then-Temporally-Refine

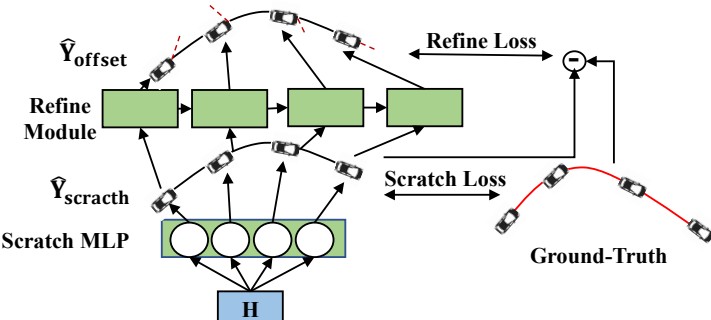

Figure 2: The proposed scratch-then-temporally-refine paradigm. An MLP is first trained to generate the scratch trajectory $\hat{\mathbf{Y}}_{\text{scratch}}$. Taking $\hat{\mathbf{Y}}_{\text{scratch}}$ as the input, the refinement module predicts the offset for each time-step. It could be a structure with temporal inductive bias (RNN/1D-CNN).

Through Section 4.1, we find that directly utilizing RNN to predict trajectories is hard because it has to generate in an autoregressive fashion, which is troublesome. Thus, to utilize the RNN while avoiding the autoregressive part, we need to generate proper input states to RNN for each time-step. We notice the fact that a simple MLP is proven to be able to do the prediction accurately in the existing literature, which inspires us to **train an MLP to generate scratch trajectories and then using structures with temporal prior (RNN or 1D-CNN) to refine the scratch.**

Formally, given the hidden feature $\mathbf{H}$ from the encoder, we first train an MLP with the output shape of $T \times 2$, denoted as $\phi_{\text{scratch}}$:

$$\{\hat{\mathbf{Y}}^t_{\text{scratch}}|t = 1,...T\} = \phi_{\text{scratch}}(\mathbf{H}). \tag{6}$$

Based on the scratch trajectories $\{\hat{\mathbf{Y}}^t_{\text{scratch}}|t = 1,...T\}$ as well as the feature $\mathbf{H}$ from the encoder which encodes agents interactions and map information, we train a refinement module $\phi_{\text{refine}}$ to refine them:

$$\{\hat{\mathbf{Y}}^t_{\text{offset}}|t = 1,...T\} = \phi_{\text{refine}}(\{\hat{\mathbf{Y}}^t_{\text{scratch}}|t = 1,...T\}, \mathbf{H}), \tag{7}$$

where the objective for $\mathbf{Y}^t_{\text{offset}}$ is the difference between $\hat{\mathbf{Y}}^t_{\text{scratch}}$ and the ground-truth $\mathbf{Y}^t$. Figure 2 gives an illustration of the proposed paradigm.

Note that $\phi_{\text{scratch}}$ and $\phi_{\text{refine}}$ could be trained in an end-to-end fashion. In experiments, we have different choices for $\phi_{\text{refine}}$: a) another MLP, similar to [3], which takes the flattened scratch trajectory as the input and output a $T \times 2$ tensor. b) an LSTM which takes the scratch trajectory as the input of the $t^{th}$ time-step . c) 1D-CNN, which serves as the local temporal information filter. The results are shown in Table 2. Additionally, for the variant with best performance, 1D-CNN, we also conduct experiments

Table 2: Performance comparison of different refinement modules. 5 * 1D-CNN means 5 cascaded 1D-CNN refinement modules.

| $\phi_{\text{refine}}$ | minADE↓ | minFDE↓ | MR↓ |
|---|---|---|---|
| None | 0.6056 | 1.2328 | 0.1723 |
| MLP | 0.6004 | 1.2300 | 0.1680 |
| RNN | 0.5931 | 1.2195 | 0.1637 |
| 1D-CNN | 0.5908 | 1.2084 | 0.1625 |
| 5 * 1D-CNN | **0.5871** | **1.1893** | **0.1554** |

on cascading multiple refinement modules to explore the effects of the multi-granularity refinement.

We can conclude that: a) An additional refinement module could improve the performance notably, even if it is simply another MLP. b) Refinement modules with temporal inductive bias (RNN and 1D-CNN) perform better than the one without such bias. c) By cascading multiple refinement modules, the performance could be further boosted.

In summary, we find that the proposed scratch-then-temporally-refine paradigm could combine the best of two worlds: the easy optimization of the MLP and the temporal inductive bias of RNN/1D-CNN.

# 5 Temporally Correlated Objective Function

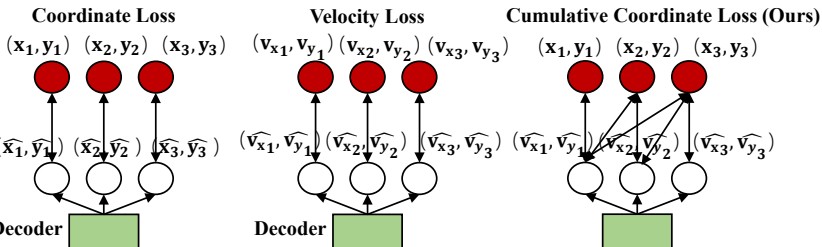

Figure 3: Three objective functions for regression. Cumulative coordinate loss achieves the best performance.

In addition to designing the neural network with the temporal structure, the temporal priors among future time-steps could be applied on the objective function part as well. Existing works [1, 6, 30, 5] usually calculate the regression loss per time-step and then take the mean of all time-steps as the overall loss:

$$\mathcal{L}(\{\hat{\mathbf{Y}}^t | t = 1, ..., T\}, \{\mathbf{Y}^t | t = 1, ..., T\}) = \frac{1}{T} \sum_{t=1}^{T} \mathcal{L}_{\text{step}}(\hat{\mathbf{Y}}^t, \mathbf{Y}^t), \tag{8}$$

where $\hat{\mathbf{Y}}^t$ and $\mathbf{Y}^t$ are the predicted and ground-truth coordinate at time-step $t$ respectively, and $\mathcal{L}_{\text{step}}$ is the loss function applied on each time-step, which could either be the MSE loss [1, 7] or NLL loss [6, 5]. This objective function assumes that agents' future positions are independent with each other, which ignores the physical constraint and could result in kinematically infeasible prediction. Additionally, for the long-term trajectory prediction, agents might move hundreds of meters in the predicted interval. As a result, the magnitude of the late time-steps' coordinates might be much larger than the early ones, which brings difficulty for the learning process.

One intuitive solution for the aforementiond issues is to predict the control signal instead [32, 31, 43] and the coordinate could be obtained by physical models during inference. In this way, the network explicitly learns the dynamics which provides strong prior knowledge to the model while keeping the output magnitude of different time-steps similar. However, in our empirical experiments, predicting velocity leads to performance drop, which is consistent with the conclusion in [5]. One burden for predicting the control signal is the cumulative error which makes the model sensitive to noise. With a prediction error occurring at the first time-step, the entire generated trajectory would shift from the ground-truth, even if predictions at all other time-steps are correct.

To alleviate the problem, we propose to let the model output velocity while calculating the regression loss between the accumulation of the models' output and the ground-truth coordinate. Formally, given the predicted velocity $\{\widehat{\mathbf{Y}}^t | t = 1, ..., T\}$ and the GT coordinate $\{\mathbf{Y}^t | t = 1, ..., T\}$, the objective is:

Table 3: Performance comparison of different objective functions. The third is the proposed cumulative coordinate loss.

| Loss Func | minADE↓ | minFDE↓ | MR↓ |
|---|---|---|---|
| Coordinate Loss | 0.6056 | 1.2328 | 0.1723 |
| Velocity Loss | 0.6346 | 1.2590 | 0.1822 |
| Cumu. Coord. Loss | **0.5921** | **1.1851** | **0.1542** |

$$\mathcal{L}_{\text{cumulative}} = \frac{1}{T} \sum_{t=1}^{T} \mathcal{L}_{\text{step}} (\sum_{m=1}^{t} \widehat{\mathbf{Y}^m}, \mathbf{Y}^t). \tag{9}$$

In this way, we combine the best of the two worlds: the output is in the form of velocity which introduces temporal correlation among time-steps while the early time-step's influence on its later coordinates is considered by the accumulation of the loss. Figure. 3 illustrates the three objective functions mentioned above and Table. 3 gives their performance. The proposed cumulative coordinate loss function achieves the best performance.

## 6 Results on the Public Leaderboard

Here we provide results on the test set of Waymo Open Motion[3] to verify that our conclusions are not over-fitting the validation set, as shown in Table 4. We can find that by combining the proposed techniques, the SOTA encoder HDGT's performance could be further boosted and achieves competitive performance on the learderboard[4].

Table 4: Performance comparison on Waymo Open Motion leaderboard, *test set*.

| Model | minADE↓ | minFDE↓ | MR↓ |
|---|---|---|---|
| ReCoAt [44] | 0.7703 | 1.6668 | 0.2437 |
| DenseTNT [7] | 1.0387 | 1.5514 | 0.1573 |
| SceneTransformer [6] | 0.6117 | 1.2116 | 0.1564 |
| MultiPath++ [5] | 0.5557 | 1.1577 | 0.1340 |
| golfer | 0.5533 | 1.1608 | 0.1354 |
| MTRA | 0.5640 | 1.1344 | 0.1160 |
| HDGT [8] | 0.5933 | 1.2055 | 0.1511 |
| **HDGT + Ours** | 0.5703 | 1.1434 | 0.1440 |

## 7 Ablation Study

Table 5: Ablation study of the proposed techniques. Note that for the temporal refinement module, we use the best setting, 5*1D-CNN.

| Temporal Refine | Cumu. Coord. Loss | minADE↓ | minFDE↓ | MR↓ | TRI(%)↓ | UR(%)↓ | Time(s) |
|---|---|---|---|---|---|---|---|
| ✗ | ✗ | 0.6056 | 1.2328 | 0.1723 | 10.95 | 2.37 | 1146 ± 16.5 |
| ✔ | ✗ | 0.5871 | 1.1893 | 0.1554 | **5.27** | **0.38** | 1181 ± 14.49 |
| ✗ | ✔ | 0.5921 | 1.1851 | 0.1548 | 12.05 | 4.87 | 1153 ± 15.33 |
| ✔ | ✔ | **0.5835** | **1.1833** | **0.1532** | 6.89 | 0.39 | 1206 ± 14.22 |

In this section, we study the effects of the combination of the proposed techniques. In addition to the commonly used metrics - minADE/minFDE/MR, to evaluate the physical feasibility of predicted trajectories under different settings, we calculate the Turning Radius Infeasibility (**TRI**) [5]. If the turning radius (the circumradius constituting three consecutive waypoints) along the predicted trajectories is less than a certain threshold, it is treated as a violation. We set this threshold as 3.5m - the approximate minimum turning radius threshold for a midsize sedan, following [5]. We also evaluate the smoothness of the trajectory by calculating the acceleration and jerk, *i.e.*, the second and third time derivative of the position. According to [45, 46], human drivers' maximum acceleration is around $5.0\text{m/s}^2$, and that of jerk is $2.0\text{m/s}^3$. Thus, we define the

[3]https://waymo.com/open/challenges/2022/motion-prediction/
[4]Note that some methods are with ensemble while ours is a single model.

Unsmooth Ratio (**UR**) as: if at any time-step of a trajectory, its corresponding acceleration is larger than $5.0\text{m/s}^2$ or jerk is larger than $2.0\text{m/s}^3$, it is considered as an unsmooth step.

The ablation study results are shown in Table 5. We can conclude that: both techniques could improve the distance-based metrics of the baseline model and the combination of them could further boost the performance. As for the Turning Radius Infeasibility (TRI) and Unsmooth Ratio (UR), the temporal refinement module could make generated trajectories much more smooth and physically feasible while the cumulative coordinate loss is harmful for them. We think it is because predictions of different time-steps have more information interaction in the temporal refinement module, which is helpful for their consistency. However, the cumulative coordinate loss entangles all the information by the accumulation process which might be unstable compared to direct regression.

Also, to evaluate the impact of this modification is on the inference time of the origin model, we calculcate the mean time and std of 30 runs while keeping the environment, devices, hyperparameters like batch-size the same. The computational burden of Cumulative Coordinate Loss is neglectable while the 5 cascaded refine modules takes 3-5% more inference time.

## 8   Conclusion

In this paper, we explore ways of applying the temporal priors on the generation process of trajectories. We find that intuitively using an autoregressive RNN would lead to degenerated performance compared to the commonly used MLP decoder. To combine the best of two methods, we propose a scratch-then-temporally-refine paradigm, which first generates a scratch trajectory by an MLP and then incorporates the temporal structure to refine the trajectory. We also find that directly fitting the velocity degenerates the performance, while accumulating the velocity and then still using the coordinate as the target would lead to better performance. Finally, by combining these discovered effective techniques, we boost the SOTA encoder - HDGT's performance by a large margin in the Waymo Motion Leaderboard, which verifies the benefits of utilizing our proposed temporal prior.

## 9   Implementation Details

We adopt HDGT [8] as the encoder for all models with the same hyperparameters. We use all agents in the scene as a sample for the wholeness of the information while [8] uses every target agent and its surrounding 16 agents as a sample. We use max-pooling as the aggregation function for the lane node for less computational cost, which we find has few degeneration of the performance. We train all ablation models with 30 epochs for quick verification of the idea and train the final submission to the leaderboard with 120 epochs which has similar training steps with [8].

## 10   Limitation and Future Work

Experiments are only conducted on one SOTA encoder on the Waymo Open Motion Dataset (WOMP). Though it is the largest public dataset for motion prediction, there still could be potential biased conclusion. To address this issue, in the future works, we might implement more SOTA encoders and conduct experiments on more datasets to further verify the conclusion.

Autoregressive RNN, as an structure naturally with temporal prior, did not work well under the simple attempts tried in this paper. More sophisticated and well-designed strcture could be explored such as [47, 48] for better convergence and performance.

### Acknowledgments

This work was partly supported by NSFC (62206172, 62222607), Shanghai Municipal Science and Technology Major Project (2021SHZDZX0102), and Shanghai Committee of Science and Technology (21DZ1100100).

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
