# OpenReview forum: "Towards Capturing the Temporal Dynamics for Trajectory Prediction: a Coarse-to-Fine Approach"
_robot-learning.org/CoRL/2022/Conference — CoRL 2022 Poster_

### Official Review · Reviewer_YWfE · 2022-07-29

**Originality:** Very Good
**Technical Quality:** Good
**Clarity Of Presentation:** Very Good
**Impact:** 2

**Recommendation:**

Weak Accept: I recommend accepting the paper, but will not argue for my recommendation if the majority of other reviewers have a different opinion.

**Summary:**

This work examines the problem of trajectory prediction.  Since most SOTA works utilize an encoder-decoder architecture with a single MLP, temporal correlation among future time-steps is ignored and therefore infeasible trajectories may be generated.  Thus far, auto-regressive decoders have not worked well empirically for trajectory prediction, but may offer some opportunity to mitigate this temporal correlation problem.

The method proposed here learns scratch trajectories using an MLP and then refines these using an RNN.  A temporal constraint loss is also used to enforce a temporal prior on the generated trajectories.  They find that the best empirical performance is achieved using a velocity prediction and coordinate prediction loss for the final objective.

The results are quite impressive when evaluated on Waymo Open Motion and the public leader-board results are compared with.  The ablations involving temporal refinement and the coordinate losses are useful and clear.


**Issues:**

-- See above: Question on Z-forcing baseline for auto-regressive architecture.

**Quality Of The Limitations Section:**

Additional details required

**Reviewer Expertise:**

2: The reviewer is willing to defend the evaluation, but it is quite likely that the reviewer did not understand central parts of the paper

**Robotics Focus:**

Highly relevant to robotics but no hardware experiments

**Strengths And Weaknesses:**

This method is very well motivated in terms of temporal correlation and the results of it are impressive.  I think the idea of using the RNN for refinement is quite novel in this problem space, but do not consider myself an expert in trajectory prediction.  The results are impressive on the public leader board. The number of baselines compared against appears to be extremely extensive, but it is hard for me to comment on the quality here since I am not familiar with most of them.  I am surprised that performance is so poor when using an auto-regressive decoder for these tasks, and this paper appears to alleviate the problem well.

The ablations are also extensive and appear to demonstrate the importance of the additional losses and refinement using the RNN.

The paper is clearly written and well positioned with literature.  The method is concisely summarized.  I could understand most of the motivation and method without much knowledge of trajectory prediction problems.

I was wondering if Z-forcing was ever utilized for these auto-regressive methods for trajectory prediction (Goyal et al., 2017)? It should show some improvements over the original teacher forcing work for the auto-regressive architecture comparisons.  This may be a baseline to be compared with. I also assume the choice of auxiliary loss could introduce temporal constraint.



**Summary Of Recommendation:**

I am not an expert in trajectory prediction, but the experimental results seem impressive and the set of baselines compared with is extensive.  The experimental procedure appears sound and the method is extensively ablated.

I was able to clearly understand the motivation for this work, it identifies a clear and unsolved problem (the difficulty of training an auto-regressive architecture for this type of trajectory prediction and temporal correlation among these trajectories) and introduces a working, well defined solution.  I cannot comment as to the significance much.

This work was also easy to read and the method is concisely presented.

---

### Official Review · Reviewer_QD7m · 2022-07-31

**Originality:** Fair
**Technical Quality:** Fair
**Clarity Of Presentation:** Fair
**Impact:** 3

**Recommendation:**

Strong Reject: I recommend rejecting the paper and will argue for my recommendation even if other reviewers hold a different opinion.

**Summary:**

To better process the temporal relationship in decoding sequential locations in trajectory prediction, this work proposes to produce the future trajectory using a scratch-then-temporally-refine strategy and optimize the NN by calculating loss from accumulating velocity into coordinates. The method is applied to an existing trajectory framework [9]. The experimental validation is conducted by comparing the common prediction performance with SOTA methods' on the public leaderboard and some feasibility-related metrics with the baseline method [9].

**Issues:**

In addition to the main weakness, the following issues need to be addressed.
- How about the performance of directly using scratch loss with the "MLP + 5x1D-CNN" structure? It could validate the necessity of "scratch loss + refine loss."
- Please claim if there is any difference between velocity output and offset output.
- How about the TRI and UR performance of other SOTA methods? By comparing with those SOTA using MLP output, it could convince if the proposed method owns superiority in temporal-related aspects.
- How does the acceleration and turning radius of discrete points are calculated? Are the effects of lateral and longitudinal acceleration both counted and separately considered?

**Quality Of The Limitations Section:**

Additional details required

**Reviewer Expertise:**

5: The reviewer is absolutely certain that the evaluation is correct and very familiar with the relevant literature

**Robotics Focus:**

Highly relevant to robotics but no hardware experiments

**Strengths And Weaknesses:**

Strengths:
- this work focuses on the temporal relationship in future prediction, which is highly related to the feasibility of predicted trajectories and greatly affects the downstream PnC in autonomous driving. This is an essential but not well-addressed topic in the area.

Weakness:
- The paradigm design of so-called scratch-then-temporally-refine looks unreasonable. The refined layers' input comes from the MLP decoder, in which the paper argues the temporal relation would be lost. However, as the only information source in equation (7), how could the refine layers recover the temporal relation? How could the refine layers capture the effect from other aspects (interaction, map, etc.)? In this way, the design of refinement parts looks like a simple smoother at the end of the decoder.
- The loss function termed temporally correlated objective loss is nothing new. Although it claims that the NN output with the form of velocity and then the loss is calculated by accumulating the velocity output as coordinates, this is exactly the same as coordinate offset prediction which is widely used in many prediction works.
- The effect has not been fully validated. All the performance improvements (including common metrics and feasibility-related metrics) come from comparing [9] with [9]+ours, but there still exists some noticeable performance gap compared with SOTA methods.
- What's most important is that if the effectiveness of the proposed paradigm could improve different prediction frameworks, it's good as a general design. If the proposed method can only improve a specific framework (as only validated on [9] in the paper), then this could only be regarded as incremental work. Therefore, it is necessary to validate the effectiveness in different prediction frameworks.


**Summary Of Recommendation:**

I would not recommend this paper as there still exists critical issues around its main contributions.
- The proposed design of scratch-then-temporally-refine discards most encoded information and thus only acts in smoothing the MLP results.
- The loss function makes no difference with commonly used offset output.
- The experiment cannot fully validate its effectiveness and universality. All the performance gain comes from the comparison with [9], which makes it more like incremental work.
- Lastly, this paper's core point of temporal constraints looks overclaimed. Even with the proposed predict-and-refine paradigm and velocity output, it can hardly claim any temporal constraints are imposed.

---

### Official Review · Reviewer_z8DX · 2022-07-31

**Originality:** Fair
**Technical Quality:** Good
**Clarity Of Presentation:** Very Good
**Impact:** 2

**Recommendation:**

Weak Reject: I recommend rejecting the paper, but will not argue for my recommendation if the majority of other reviewers have a different opinion.

**Summary:**

Most of the current trajectory prediction methods use MLP as the final decoder, which ignores the inherent temporal correlation constraint in the future trajectory. To solve this problem, the authors propose Scratch-then-Temporally-Refine for the decoder and Temporally Correlated Objective Function for the loss to impose the temporal inductive bias during the trajectory generation. Through the experiment on the Waymo Open Motion Challenge, the authors show the improvement of the current top-ranked method by using the above methods. The ablation suggests that not only the distance-based metrics are boosted but also the feasibility e.g. Turning Radius Infeasibility (TRI) and Unsmooth Ratio (UR).

**Issues:**

* I am curious why the additional refinement module (MLP, RNN, 1D-CNN) could improve the prediction performance. Is there any intuitive explanation? Could you please add more discussions?

**Quality Of The Limitations Section:**

Additional details required

**Reviewer Expertise:**

4: The reviewer is confident but not absolutely certain that the evaluation is correct

**Robotics Focus:**

Highly relevant to robotics but no hardware experiments

**Strengths And Weaknesses:**

Strengths

* I enjoy the idea “Scratch-then-Temporally-Refine”. Since the Autoregressive RNN and MLP-based trajectory refinement are commonly used in the decoder independently, the authors combine them together to further improve prediction results. But it seems to be just a trick in the decoder.

* The ablation study is comprehensive and convincing. Detailed experiments have been conducted to indicate the superiority of each module.

* The paper is clearly written and easy to read.

Weaknesses

* The proposed loss function “Temporally Correlated Objective Function” is not novel. Concretely, the prediction header generates control signal velocity at each time step, then the predicted location is computed by velocity accumulation. This process is identical with Producing Dynamically-Feasible Trajectories in the paper [“Trajectron++: Dynamically-Feasible Trajectory Forecasting With Heterogeneous Data”](https://www.ecva.net/papers/eccv_2020/papers_ECCV/papers/123630664.pdf).

* As mentioned by authors in paper, more SOTA encoder should be considered to verify the performance of the proposed methods.

**Summary Of Recommendation:**

Although the “Scratch-then-Temporally-Refine” module shows promising performance for the decoder, the “Temporally Correlated Objective Function” loss has been used in the previous work. Thus, the overall contribution is not strong enough to support paper publishing at CoRL.

---

### Official Review · Reviewer_G9vs · 2022-08-03

**Originality:** Very Good
**Technical Quality:** Excellent
**Clarity Of Presentation:** Excellent
**Impact:** 4

**Recommendation:**

Strong Accept: I recommend accepting the paper and will argue for my recommendation even if other reviewers hold a different opinion.

**Summary:**

The paper analyses why current SotA motion forecasting methods use MLP output heads and not recurrent/auto-regressive ones. The paper then proposes several improvements to an MLP-output method that brings in benefits from RNN-based approaches. This leads to a noticeable score improvement on the public leaderboard for the WOMD challenge.



**Issues:**

I already think the paper is in a very good spot and could definitely be accepted but I'd appreciate it if the authors could respond to points **W.1** and **W.3** above. In particular, a standalone Jupyter notebook would really make me excited.

**Quality Of The Limitations Section:**

Limitations are addressed clearly

**Reviewer Expertise:**

4: The reviewer is confident but not absolutely certain that the evaluation is correct

**Robotics Focus:**

Relevant but unlikely to deploy to hardware in near future

**Strengths And Weaknesses:**

This paper was such a fun read. The structure is more like a student report than a classic method-experiments-results paper but that works in its favor. The authors make a very convincing case that there is a problem with using RNNs to decode waypoints from the latent. After that, the authors carry out meaningful experiments that lead to a clear improvement of a method on the official leaderboard.

### Strengths

- **S.1** Clarity. The writing is clear and concise. The structure is great. All method parts are very easy to follow. The figures are excellent and help greatly in making the point. I can't stress this point enough. Among my 5 review assignments this CoRL, this was the best read.
- **S.2** Results. The authors submitted their results to the official WOMD leaderboard and I verified that the results match the ones reported in the paper. The improvement of 0.02 points is very notable, as this has increased the leaderboard rank of the method by one and reduced the margin to the next-best method down to 0.006.
- **S.3** I appreciate that the negative results of using RNNs on the HDGT method (sec. 4.1) were included. It serves as a great jumping-off point for the rest of the paper.
- **S.4** Limitation section is honest and spot-on.

### Weaknesses

- **W.1** Reproducibility. I wished you would've included code. A standalone Jupyter notebook on a toy example would be great for making this method accessible to others.
- **W.2** Generality. This is already discussed in your Limitations section and I agree - the paper would be stronger with more different backends and applied to more different datasets.
- **W.3** Robotics focus. Since this is CoRL, we reviewers are asked to mark how relevant to real-world robot experiments this method is, and right now I don't see any mention of deployment. I'd be curious what the impact of this modification is on the inference time of HDGT.

**Summary Of Recommendation:**

I vouch for this paper. The method is well-justified and achieves good results on a famous public benchmark.

---

### Meta-Review · Area_Chair_jAA7 · 2022-08-05

**Recommendation:** Accept (Poster)
**Confidence:** 4

**Metareview:**

This paper explores mechanisms to include temporal information in trajectory prediction. The paper highlights that in general approaches that attempt to include temporal information in the output stage of a network perform more poorly than an MLP output head, but this seems to be contradictory, given that trajectories do exhibit strong temporal information. The authors investigate the use of a residual output head that predicts trajectories using an MLP, alongside an additional temporal refinement module. The head is trained using a loss on both the MLP and refined head, and explored along with a number of other ideas shown to outperform existing techniques on a number of datasets.

Reviewers highlighted the well organised paper and thorough investigation of the impacts of different model heads, alongside impressive results when these techniques were combined. However, reviewers were extremely polarised by this paper, and there has been extensive debate about the strength of the contribution, in particular because the work does not contribute a particularly novel new approach, rather provides a study of existing techniques and builds on these. Of the key improvement methods studied (scratch+refine & temporally correlated objective function) here, reviewers felt that only the first is a novel contribution, as the second is relatively widely used and understood to be an effective choice.

There were some concerns about the refinement network not conditioned on latent state, which I think this have been addressed in the revised version with an updated equation that shows that the architecture allows more than just smoothing. Additional ablations provided by the authors in response to the reviewers corroborated initial findings, and improve the paper.

While I understand that this work may not take the form of a typical paper, as it provides a thorough study of predominantly pre-existing techniques rather than introducing an entirely new architecture or method, I still believe that the findings in this work are useful, and that the scientific merit of this work outweighs these concerns. I appreciate this form of analysis, and would like to see more of this in our community. As a result, I am recommending this work be accepted.

The primary area in need of addressing for the camera ready paper relates to a lack of rigour in places (eg. the equation description missing latent variables), and loose/ imprecise language and choice of wording that leads to confusion (eg. referring to a temporal constraint, when there is no such constraint enforced). I would recommend that the authors carefully proof read the paper, and consider changing the title to use more precise wording (eg. structure instead of constraint).